# Central insulin modulates food valuation via mesolimbic pathways

Lena J. Tiedemann[1], Sebastian M. Schmid[2,3], Judith Hettel[1], Katrin Giesen[1], Paul Francke[1], Christian Büchel[1] & Stefanie Brassen[1]

Central insulin is thought to act at the neural interface between metabolic and hedonic drives to eat. Here, using pharmacological fMRI, we show that intranasal insulin (INI) changes the value of food cues through modulation of mesolimbic pathways. Overnight fasted participants rated the palatability of food pictures and attractiveness of non-food items (control) after receiving INI or placebo. We report that INI reduces ratings of food palatability and value signals in mesolimbic regions in individuals with normal insulin sensitivity. Connectivity analyses reveal insulinergic inhibition of forward projections from the ventral tegmentum to the nucleus accumbens. Importantly, the strength of this modulation predicts decrease of palatability ratings, directly linking neural findings to behaviour. In insulin-resistant participants however, we observe reduced food values and aberrant central insulin action. These data demonstrate how central insulin modulates the cross-talk between homeostatic and non-homeostatic feeding systems, suggesting that dysfunctions of these neural interactions may promote metabolic disorders.

[1] Department of Systems Neuroscience, University Medical Centre Hamburg-Eppendorf, Martinistrasse 52, D-20246 Hamburg, Germany. [2] Department of Internal Medicine I, University Hospital Lübeck, Ratzeburger Allee 160, D-23538 Lübeck, Germany. [3] German Center for Diabetes Research (DZD), Ratzeburger Allee 160, D-23538 Lübeck, Germany. Correspondence and requests for materials should be addressed to S.B. (email: sbrassen@uke.de).

In light of the dramatically rising incidence of obesity and type 2 diabetes (T2D), both of which are related to the overeating of highly palatable foods, it is critical to understand the neural control of ingestive behaviour. Metabolic and hedonic feeding is driven by specific homeostatic and non-homeostatic neurocircuits[1] and insulin, which is a key effector hormone of energy balance, signals in both systems. Overeating creates a continuous supply of energy that in turn leads to perpetually elevated circulating levels of insulin and insulin resistance (IR)[2–4]. Thus, investigating how baseline measures of central insulin can change the neural control of food processing in non-homeostatic pathways may substantially enhance our understanding of the neural cross-talk between homeostatic and reward-related feeding systems and how dysfunctions in this cross-talk may promote pathological eating behaviour.

Findings in rodents indicate that, apart from signalling in hypothalamic neurocircuits regulating energy homeostasis, central insulin mediates non-homeostatic feeding for pleasure by signalling within mesolimbic reward circuits[2,5,6]. The mesolimbic pathway is thought to critically mediate different aspects of reward processing[7,8] and insulin receptors are expressed throughout these brain regions[5,9]. Accordingly, direct injections of insulin into the ventral tegmental area (VTA)[10,11] and in the nucleus accumbens (NAc)[12] impact dopamine (DA) release. For example, direct administration of insulin into the VTA reduces hedonic feeding under sated conditions and depresses somatodendritic DA in the VTA. Insulin-induced depression of somatodendritic DA has been attributed to the upregulation of the number or function of DA transporter in the VTA[11]. Moreover, insulin injection decreases glutamatergic synaptic transmission (long-term depression, LTD) onto VTA DA neurons, which in turn may reduce DA burst activity and subsequent DA release in mesocorticolimbic regions[10]. Given strong reciprocal connections between the VTA and the NAc, which encodes the subjective value of rewards[7], insulin-mediated depression of DA activity in the VTA might suppress salience of food through reduced DA release in the NAc. According to this hypothesis, central insulin action has been connected to depressed hedonic feeding, reduced food anticipatory behaviour and lower preference for food cues in animals[10,11,13], even though direct food-related responses in the NAc following VTA modulation have not been recorded and findings are restricted to rodents.

Over time, overconsumption of energy-dense diets and the associated gain in body weight result in decreased peripheral and central insulin sensitivity, and elevated concentrations of circulating insulin[14]. Selective effects of aberrant central insulin action in humans, however, are under debate[4] and even unstudied when it comes to reward-related networks. Neuroimaging studies in obese individuals have reported conflicting findings, ranging from hyper- to hypoactive neural responses to food stimuli[15]. One reason for this apparent controversy might be the uncontrolled impact of neuroendocrine signals, such as insulin, on food processing. For example, the frequently discussed reward deficiency theory in obesity is based on observations of decreased striatal signals in obesity[16–18]. Such a hypofunction has predominantly been attributed to pre-existing vulnerability in the dopaminergic system as well as to adaptive neuroplasticity following perpetual overeating[17,19]. The potential role of elevated central insulin levels on the regulation of food processing in hyperinsulinemic humans, however, has not yet been studied. In this context, interesting findings in rodents have demonstrated that exposure to sweetened, high-fat food induces synaptic depression onto DA neurons[10], and that insulin-mediated LTD of VTA DA neurons is reduced in hyperinsulinemia[20].

To elucidate the direct impact of central insulin on the preference for food-related cues and VTA-NAc circuits in humans, we combined pharmacological functional magnetic resonance imaging (fMRI) with a food/non-food valuation paradigm done by participants who fasted the previous night. We studied participants with normal insulin sensitivity as well as non-diabetic individuals with IR, who are at risk for T2D[21,22], to investigate central insulin effects under physiological and pathological circumstances. Using a placebo-controlled double-blind crossover design, we investigated the effects of central insulin by making use of the intranasal route of insulin administration (INI). INI application in humans has been shown to bypass the blood–brain–barrier and effectively deliver insulin to the central nervous system (CNS) within 30 min after administration in the absence of relevant systemic absorption[23,24]. This approach allows us to ensure that our findings in individuals with reduced whole-body insulin sensitivity are not confounded by potentially attenuated transport of the hormone across the blood–brain–barrier[25].

Our findings show that INI specifically reduces preference ratings for food-cues and suppresses food-value signals in the NAc by negatively modulating projections from the VTA in individuals with normal insulin sensitivity. In contrast, insulin-resistant participants (at risk for T2D) show reduced neurobehavioural food valuation at baseline as well as after INI, indicating a critical role of central insulin action in mesolimbic pathways for the processing of food value and salience in the human brain.

## Results

**Task overview.** After an overnight fast of at least 10 h ($12.8 \pm 1.2$ h), all participants underwent a 2-day fMRI scanning procedure, separated by at least 1 week ($8.7 \pm 3.8$ days) that was combined with 160 IU INI or placebo in a double-blind, randomized crossover design (Fig. 1a). In the scanner, participants were asked to rate the overall preference for food and non-food items with yes ($\sim$ 'I like this') or no ($\sim$ 'I do not like this') by button press, which was followed by a four-point rating scale where they were asked to provide a detailed rating, indicating how much they liked or disliked each item (Fig. 1b and Supplementary Fig. 1). Stimuli were presented in pseudo-randomized order. Parametric values were derived from transferring the general and the four-point rating into a single scale ranging from 1 ('not at all') to 8 ('very much'). It is noteworthy that stimulus sets of both days were comparable regarding picture salience and likability as ensured by an independent validation study (see Supplementary Table 1).

**Insulin groups.** Forty-eight normal to overweight non-diabetic volunteers participated in the study and were classified into insulin groups based on insulin sensitivity as defined by the well-established homeostatic model assessment using a cut-off of < 2.0 (Homeostatic Model Assessment for Insulin Resistance (HOMA-IR)[26]). Normal insulin sensitivity was identified in $n = 28$ participants (normal insulin resistance (NIR) 14 male), whereas $n = 20$ individuals fulfilled criteria for IR (9 male). Normal HbA1C values confirm the exclusion of diabetes in our insulin-resistant participants who are at risk for T2D but in whom elevated insulin release may still compensate for reduced insulin sensitivity (Table 1).

General preference for different kind of foods was comparable in both groups (Supplementary Table 2). Gender, age, overnight fasting time (Supplementary Table 3a), days between scanning sessions and hunger ratings also did not differ between groups (all $P > 0.33$, $n_{NIR} = 28$, $n_{IR} = 20$, t-test). As expected, individuals in the IR group demonstrated enhanced scores in all body measurements and adiposity-related blood values before scanning, that is, showed elevated levels of leptin, c-peptide, insulin and glucose. Fasting glucose levels confirmed fasting state in all participants. Additional analyses on the caloric content of the protocolled last meal before fasting in each participant

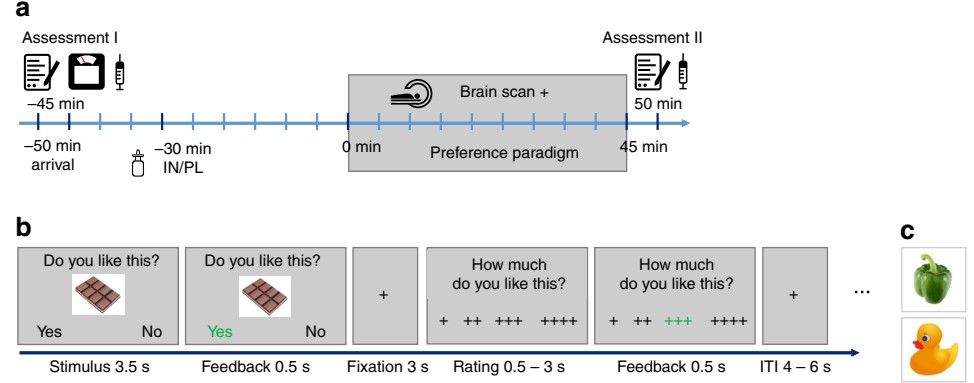

**Figure 1 | Outline of study design and experimental task.** (**a**) Experimental protocol. (**b**) Schematic representation and timing of the experimental paradigm. On each trial, a food or non-food picture (pseudo-randomized) was presented for 4 s. During the first 3.5 s, participants had to indicate the general liking of the depicted item, by pressing one of two buttons. Feedback of the chosen answer was provided for 0.5 s. After a fixation period of 3 s and during a maximum duration of 3 s, participants were asked to detail their preference rating, that is, how much they like (this example) or dislike (see Supplementary Fig. 1) the item using a four-point rating scale, by pressing one of four buttons. After another feedback of 0.5 s, the trial ended with a random fixation period of 4–6 s. (**c**) Examples of less palatable food and non-food stimuli.

**Table 1 | Sample characteristics.**

|  | NIR ($n = 28$) | IR ($n = 20$) | *P* |
|---|---|---|---|
| Age | 25.7 (0.7) | 26.1 (0.7) | NS |
| Sex (female/male) | 14/14 | 11/9 | NS |
| BMI (kg m$^{-2}$) | 23.6 (0.7) | 29.4 (1.1) | *** |
| Waist (cm) | 78.3 (1.9) | 88.3 (2.3) | ** |
| Body fat (%) | 22.6 (1.5) | 32.0 (2.0) | ** |
| Blood |  |  |  |
| HOMA-IR | 1.2 (0.1) | 2.4 (0.2) | *** |
| Glucose (mmol l$^{-1}$) | 4.7 (0.1) | 4.9 (0.1) | * |
| Insulin (pmol l$^{-1}$) | 41.0 (2.5) | 77.8 (4.8) | *** |
| Leptin (µg l$^{-1}$) | 4.9 (0.9) | 14.7 (3.1) | ** |
| C-peptide (nmol l$^{-1}$) | 0.5 (0.04) | 0.7 (0.02) | *** |
| Cortisol (nmol l$^{-1}$) | 48.2 (6.5) | 61.0 (6.6) | NS |
| HbA1C (%) | 4.9 (0.1) | 4.9 (0.1) | NS |

BMI, body mass index; HOMA-IR, Homeostatic Model Assessment for Insulin Resistance; IR, insulin resistance; NS, not significant.
***$P < .001$, **$P < .01$ and *$P < .05$, s.e.m. in parentheses.

demonstrated no difference in caloric intake between sessions, groups and regarding group × session interactions (Supplementary Table 3b).

Placebo and insulin sessions did not differ across individuals with respect to pre-scan insulin, glucose, cortisol, c-peptide, leptin, hunger ratings and time fasted (all $P > 0.18$, $n = 48$, t-test), nor were there any group × session differences in these parameters (all $P > 0.16$, $n_{NIR} = 28$, $n_{IR} = 20$, repeated measures analysis of variance (rmANOVA)). Similarly, changes in pre- compared with post-scan hunger ratings, as well as levels of glucose, did not differ between the placebo and insulin session across and between groups (all $P > 0.14$, $n_{NIR} = 28$, $n_{IR} = 20$, rmANOVA). As expected, plasma insulin levels across all participants decreased over time ($F_{(1,46)} = 8.16$; $P = 0.006$, $\eta^2 = 0.15$, $n = 48$, rmANOVA) and this decrease across individuals did not differ between the insulin and the placebo session ($F_{(1,46)} = 1.25$; $P = 0.27$, $n = 48$, rmANOVA). There was however a significant group by session interaction ($F_{(1,46)} = 4.44$; $P = 0.04$, $\eta^2 = 0.09$, $n_{NIR} = 28$, $n_{IR} = 20$, rmANOVA), which mainly was driven by a stronger plasma insulin decline at the placebo compared with the insulin session in NIR individuals (see Supplementary Table 4).

**Food item preference is decreased in IR**. To characterize baseline conditions, we first analysed data from the placebo session. In both groups, food items were liked significantly more than non-food items on the categorical and the parametric level (all $F_{(1,46)} > 110$; all $P < 0.001$, all $\eta^2 > 0.71$, $n_{NIR} = 28$, $n_{IR} = 20$, rmANOVA). NIR participants, compared with IR individuals, more frequently reported preference (that is, responded 'yes') for food (relative to non-food) items compared with IR individuals ($F_{(1,46)} = 5.49$; $P = 0.02$, $\eta^2 = 0.12$, $n_{NIR} = 28$, $n_{IR} = 20$, rmANOVA) (Fig. 2a). This reduced food preference in IR participants in the placebo session was also reflected in a trend towards reduced parametric food preference scores in IR compared with NIR participants ($F_{(1,46)} = 3.34$; $P = 0.07$, $\eta^2 = 0.07$, $n_{NIR} = 28$, $n_{IR} = 20$, rmANOVA) (Fig. 2b).

Next, we were interested in whether plasma insulin levels obtained after scanning were directly related to reported food preference values. Here, insulin levels were correlated with food preference scores only in NIR individuals: those individuals with higher insulin concentrations reported lower preference for food items ($r = -0.43$; $P = 0.02$, $n = 28$, Pearson's correlation). This correlation differed significantly from that observed in the IR group (Fisher's $Z = 2.02$; $P = 0.02$, Cohen's $q = 0.63$, $n_{NIR} = 28$,

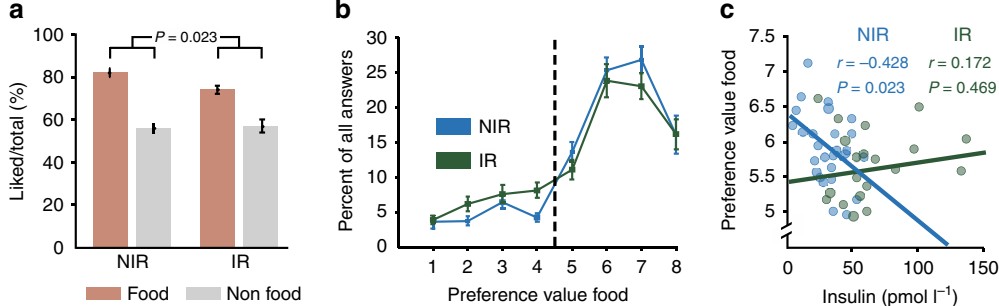

**Figure 2 | Behavioural results in the placebo condition.** (**a**) Groups means and s.e.m. of percentage of liked food and non-food items during placebo demonstrate reduced food value scores in IR. (**b**) Group means and s.e.m. of combined preference values ranging from 1 ('not at all') to 8 ('very much'). The dashed line separates 'yes' from 'no' decisions. (**c**) Correlation between individual post-scan plasma insulin levels and preference values for food items during placebo. Only in NIR plasma insulin predicted preference values.

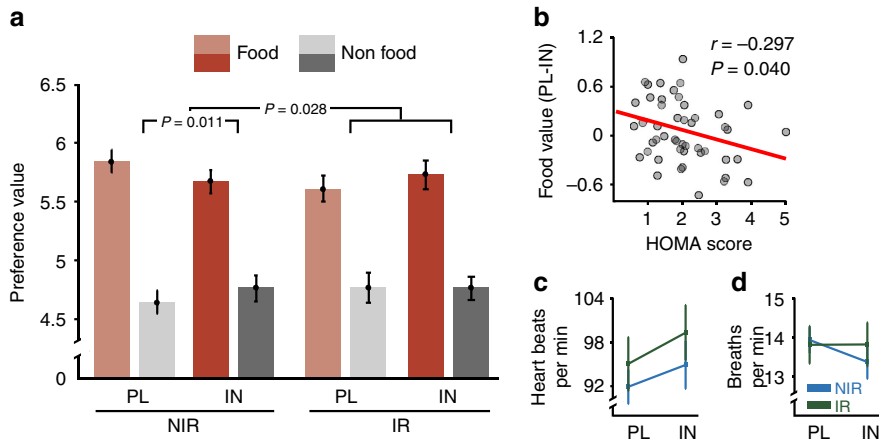

**Figure 3 | Behavioural insulin effects and autonomic data.** (**a**) Group means and s.e.m. of preference values for food and non-food stimuli during placebo (PL) and insulin (IN). rmANOVA revealed significantly reduced preference values specifically of food items under INI only in NIR, whereas food values tend to increase in IR. (**b**) INI-mediated changes in food preference scores (placebo session minus insulin session) were directly correlated to individual peripheral insulin sensitivity as defined by the HOMA index, across all participants. The lower the HOMA score the more food values were decreased under insulin. Group means and s.e.m. of heart rate (**c**) and respiration (**d**) recorded during the placebo and the insulin session.

$n_{IR} = 20$) in whom no significant relation emerged ($r = 0.17$; $P = 0.47$, $n = 20$, Pearson's correlation) (Fig. 2c).

**Increase in central insulin reduces food values only in NIR.** To assess insulin-mediated effects on parametric preference ratings, we used a rmANOVA on preference ratings and included the following factors: item (food/non-food), session (placebo/insulin) and group (NIR/IR). Results revealed a significant three-way-interaction ($F_{(1,46)} = 5.13$; $P = 0.028$, $\eta^2 = 0.1$, $n_{NIR} = 28$, $n_{IR} = 20$, rmANOVA). Preference ratings of food compared with non-food items were significantly reduced after INI application compared with placebo in the NIR group only ($F_{(1,27)} = 7.37$; $P = 0.011$, $\eta^2 = 0.22$, $n = 28$, rmANOVA). Among individuals with normal insulin functioning, INI reduced the preference of food items ($T_{(27)} = 2.31$; $P = 0.03$, $d = 0.32$, $n = 28$, t-test) but had no significant impact on non-food items ($P > 0.11$, $n = 28$, t-test). In contrast, among individuals with IR, food preference scores showed a trend to increase under INI ($T_{(19)} = 1.77$; $P = 0.09$, $d = 0.16$, $n = 20$, t-test) (Fig. 3a). In agreement with this observation, maximal HOMA scores were directly correlated with insulin-mediated changes in food preference scores, such that individuals with normal HOMA values demonstrated stronger reduction of food values

following INI application ($r = -0.30$; $P = 0.04$, $n = 48$, Pearson's correlation) (Fig. 3b).

To rule out that observed changes were primarily driven by differences in body weight we re-ran the behavioural analyses including body mass index (BMI) as a covariate. Results revealed no significant impact of BMI on observed changes, that is, group interactions in insulin effects remained significant ($F_{(1,45)} = 6.50$; $P = 0.014$, $\eta^2 = 0.13$, $n_{NIR} = 28$, $n_{IR} = 20$, rmANOVA). Moreover, there was no significant correlation between BMI and insulin-mediated changes in food preference scores across participants ($r = -0.11$; $P = 0.46$, $n = 48$, Pearson correlation).

We then investigated whether potential changes in plasma insulin during the insulin session were directly associated with observed behavioural insulin effects (that is, score changes in food liking) and found no significant correlations across and within groups (all $P > 0.21$, $n = 48$, Pearson's correlation). Interestingly, in contrast to the placebo session, post-scan insulin levels no longer explained any significant variation in food liking scores in NIR individuals ($r = -0.16$; $P = 0.41$, $n = 48$, Pearson's correlation).

During fMRI scanning, cardiac and respiratory signals were recorded and were analysed in 42 participants. rmANOVAs yielded no significant effects of group or condition on heart rate

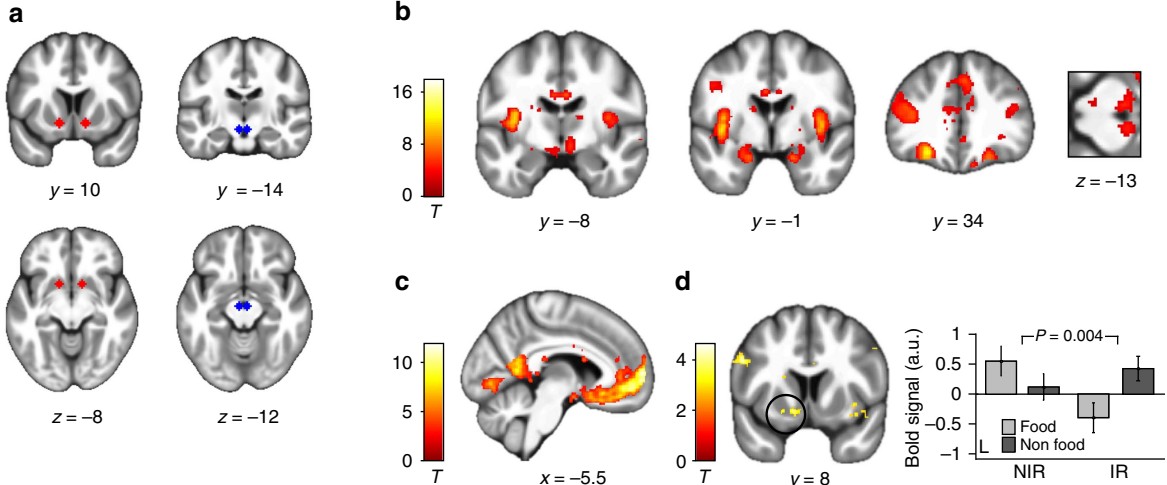

**Figure 4 | Paradigm-induced activation patterns.** (**a**) Mesolimbic ROI in the NAc (red) and the VTA (blue) overlaid on the mean structural image of all participants. (**b**) Categorical effect of food stimulus presentation. Greater activity in the insula, amygdala, orbitofrontal cortex, VTA and hypothalamus was observed in the food compared to the non-food condition across both groups. (**c**) Neural representation of preference values (parametric analysis). Regions in which the correlation with the preference value was significant in both the food and the non-food conditions across all participants included the ventromedial prefrontal cortex, posterior cingulate cortex and the bilateral NAc. (**d**) Group differences in food value signals. NIR demonstrated increased food value signals in the left NAc relative to non-food signals compared to insulin-resistant individuals (IR). Plotted contrast: $NIR_{food > non-food} > IR_{food > non-food}$. The bar graph shows group means and s.e.m. of mean parameter estimates extracted from the ROI of the left NAc. All peaks are $P < 0.05$ FWE corrected. Activations are overlaid on the mean structural image of all participants (display threshold $P < 0.005$ uncorrected).

or respiration (all $P > 0.10$, $n_{NIR} = 28$, $n_{IR} = 20$) (Fig. 3c,d and Supplementary Methods). Reaction times were significantly faster in food compared with non-food trials for all participants ($T_{(47)} = 3.00$; $P = 0.004$, $d = 0.2$, $n = 48$, t-test), but no group differences emerged ($P = 0.73$, $n_{NIR} = 28$, $n_{IR} = 20$, rmANOVA).

**Food valuation activates hedonic and metabolic neurocircuits.** To examine how INI influenced the brain's mesolimbic reward circuitry, we analysed blood oxygenation level-dependent (BOLD) activity measured during the preference task using a two-level random effects model. Subjective preference values of each decision, ranging from 1 ('no'—'not at all') to 8 ('yes'—'very much'), were included as parametric regressors of food and non-food conditions in the model. At the second level, we used a three-factorial design including the factors item (food/non-food), session (placebo/insulin) and group (NIR/IR) to address our research questions. Imaging findings were reported when passing a family-wise error (FWE) correction at the whole-brain level or within regions of interest (ROIs), that is, the NAc and the VTA (Fig. 4a; see Methods). Investigating BOLD responses to food compared with non-food items in the placebo session yielded highly significant activations across all participants in a large-scale network of metabolic and reward-related brain regions including the bilateral hypothalamus, VTA, amygdala, insula and orbitofrontal cortex (Fig. 4b and Supplementary Table 5). These activation patterns did not differ between groups.

**NAc food value signals are reduced in IR.** Next, we tested for regions that encode subjective value, that is, identified regions that show a positive correlation between the amplitude of the BOLD response and subjective preference values in both food and non-food conditions. Results revealed strong activation of a valuation network[27,28], including the ventromedial prefrontal cortex, the posterior cingulate cortex and the bilateral NAc (all $P < 0.05$ FWE corrected, $n = 48$, factorial design) (Fig. 4c). We next identified regions showing food-specific valuation signals, that is, regions in which the correlation between preference and

BOLD response was greater for the food, as compared with the non-food, condition. This analysis revealed that the bilateral NAc was specifically engaged during food value encoding in NIR individuals, but no significant activation differences emerged for participants in the IR group. Consequently, there was a significant group interaction in the left NAc ($P < 0.05$ FWE corrected, $n_{NIR} = 28$, $n_{IR} = 20$, factorial design) (Fig. 4d).

**INI reduces mesolimbic food value signals only in NIR.** We then investigated the effects of INI on value signals in these reward circuits. First, we focused on general changes in neural value signals and compared parametric activation patterns evoked by both the food and non-food conditions in the placebo session with those patterns evoked in the insulin session. Analyses across and between groups yielded no significant changes. We next analysed central insulin effects on food-specific valuation responses (food > non-food) and found a significant group interaction in the NAc (peak left: $-12$, 8, $-8$, $P = 0.014$ FWE corrected and peak right: 10, 8, $-7$, $P = 0.046$ FWE corrected, $n_{NIR} = 28$, $n_{IR} = 20$, factorial design; Fig. 5a) and the left VTA (peak left: $-4$, $-12$, $-14$, $P = 0.042$ FWE corrected and peak right: 4, $-12$, $-14$, $P = .078$ FWE corrected; Fig. 5b). This indicates that although INI reduced the food-specific valuation signal in NIR individuals in these regions, this signal increased in IR individuals.

**Dynamic causal modelling.** Building upon these results, we wanted to know whether the observed functional neural changes could be explained by changes in connectivity. Based on recent rodent data[29], we were especially interested in whether central insulin modulates forward, backward or both (that is, bidirectional) projections between the VTA and the NAc. To this end, we used dynamic causal modelling (DCM) on adjusted BOLD time series from the VTA and the NAc. As the general linear model (GLM) results were particularly pronounced in the left hemisphere, we initially focused on the left VTA and the left NAc. Results for the right hemisphere were very similar

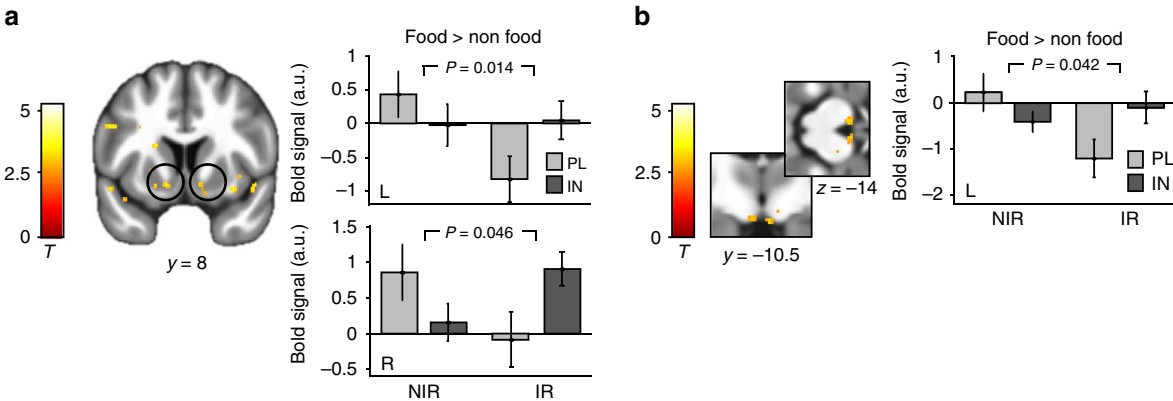

**Figure 5 | Neural insulin effects.** General linear modelling of value signals from the placebo and the insulin session revealed group by session interactions in the bilateral NAc (**a**) and the left VTA (**b**). In both regions, food value signals were decreased in the insulin session only in NIR, while signals were increased under INI in IR. Bar plots show group means and s.e.m. of mean contrast estimates extracted from ROIs from the comparison NIR_{PL > IN} > IR_{PL > IN}. Displayed P-values are FWE corrected for bilateral masks. Activations are overlaid on the mean structural image of all participants (display threshold P < 0.005 uncorrected).

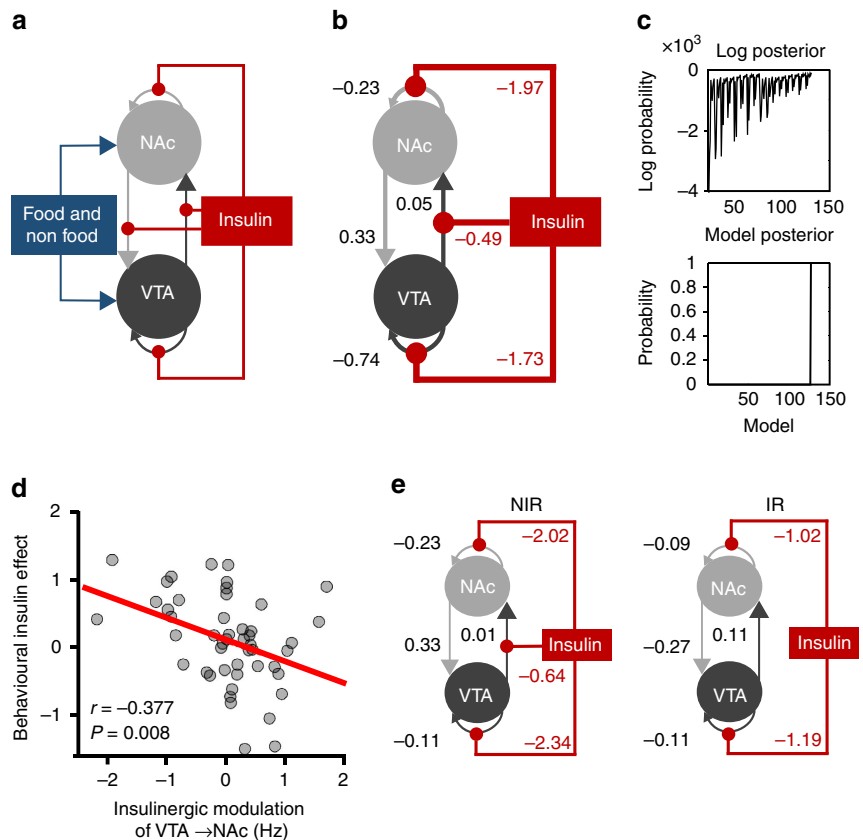

**Figure 6 | DCM results.** (**a**) The full model that was defined and inverted for each participant included all (intrinsic) self-connections and (extrinsic) forward and backward connections, food and non-food stimuli as driving inputs, and insulin modulation of all extrinsic and intrinsic connections. (**b**) The winning model: the reduced model with the highest evidence across all participants as identified through *post-hoc* optimization includes modulation of the VTA-NAc forward connection and of self-connections by insulin. Data are given as Bayesian parameter averages. These parameters indicate positive connection strength between the VTA and the NAc, and insulin to selectively inhibit forward VTA-NAc connections as well as intrinsic self-connections of both regions. (**c**) The upper graph shows the range of log-posterior probabilities of all possible models examined in the left hemisphere. The lower graph shows the posterior probability of the reduced model which had the posterior probability of (almost) 1 suggesting that the reduced model had more evidence than any other variant. The next most probable model's probability was very low (almost 0, the log-probability was − 76.88). (**d**) Correlation between neural and behavioural insulin effects. Individual parameter estimates of INI modulation on the forward connection from the VTA to the NAc correlated with insulin-mediated changes of preference scores for food relative to non-food items (Δ insulin effect food − non-food). Inhibitory modulation predicted stronger decrease of food values under INI across all participants. (**e**) The winning model selected for the different groups: within IR, there was no significant modulation by INI of the VTA to NAc forward connection.

and are presented in Supplementary Table 6. Using model selection[30] (see Methods), a full model was defined and inverted for each participant that included all potential modulatory insulin inputs on VTA-NAc connectivity (Fig. 6a).

Bayesian model reduction identified the model with the best evidence by comparing the evidence for all possible 2,048 models (Fig. 6c). The winning model included reciprocal positive connections between the VTA and the NAc, as well as negative intrinsic connections of both regions (Fig. 6b,c). Most importantly, the model also included negative modulation by insulin of the forward projections from the VTA to the NAc but no modulatory effect on backward projections. The model also indicated a negative modulation of the self-connection of both regions by insulin. Using one-sample $t$-tests, we confirmed that each of the four parameters quantifying (self-)connections was significantly different from zero, which is consistent with the expectation that these two mesolimbic subregions are strongly connected[7,31]. Similarly, parameters reflecting insulinergic modulation were significantly different from zero (all $P < .007$ Bonferroni-corrected for multiple comparisons).

Next, we tested whether behavioural effects (preference ratings) are mediated by mesolimbic connectivity. Therefore, we tested whether individual modulatory parameters of the winning model are directly associated with food-specific value reduction under insulin $((food–non-food)_{placebo} − (food–non-food)_{insulin})$. This correlational analysis revealed a significant negative association ($r = −0.38$; $P < 0.008$, $n = 48$, Pearson's correlation), indicating that negative modulation of the forward connection from the VTA to the NAc predicted food-specific value reduction under INI (Fig. 6d). Behavioural findings did not correlate with pharmacological effects on intrinsic connections ($P > 0.33$, $n = 48$, Pearson's correlation). These results suggest that a decrease in food value was related to a reduced NAc drive from the VTA by stimulation of insulin receptors.

Finally, we were interested in whether the results from the Bayesian model selection differ between NIR and IR individuals. For that purpose, we re-ran *post-hoc* model selection for the two groups separately. The identified winning model for the NIR group was identical to the selected model across all participants. In the IR group, however, modulation of extrinsic connections by insulin was no longer selected to explain neural activity in mesolimbic ROIs during our task. Nevertheless, intrinsic self-connections of both regions were still negatively modulated by INI in IR individuals (Fig. 6e).

For the sake of completeness, we repeated all analyses for the right hemisphere. Selected models across and between groups were almost identical to those of the left hemisphere (see Supplementary Table 6).

## Discussion

Our findings reveal distinct patterns of central insulin effects on behaviour and brain activity in individuals with and without IR. In healthy volunteers with normal fasting insulin levels, INI specifically decreases food palatability ratings. Reduced valuation of food palatability is directly associated with decreased food-value signals in the VTA and the NAc. These findings validate and extend work in animal models that demonstrated insulin-mediated depression of DA activity in the VTA paralleled by decreased salience of food stimuli[10,11,13]. For example, insulin injection in rodents induced LTD of excitatory synapses onto VTA DA neurons, most likely attenuating DA release in the mesocorticolimbic DA system and selectively reducing the preference for contextual cues associated with food reinforcement, as measured by conditioned place preference[10]. Insulin-mediated behavioural changes, as described in these

studies, rely on alterations in the subjective valuation of stimuli; this valuation is encoded in striatal subregions[7]. Insulin effects on the cross-talk between midbrain and ventral striatal systems, however, have not yet been studied until now. Here, we demonstrate that INI reduces NAc food-value signals. Moreover, using DCM, we can show that variation in the NAc BOLD signal is mediated by insulinergic modulation of the extrinsic forward connection from the VTA to the NAc. As the positive connection from the VTA to the NAc is inhibited under INI, dopaminergic drivers of NAc value signals probably decrease in the insulin condition. Accordingly, the individual degree of insulinergic inhibition of VTA-NAc connections directly predicts the degree of behavioural food devaluation, following INI across all participants. Recent optogenetic findings identified reward and feeding-specific circuits in which metabolic signals from the lateral hypothalamus disinhibit VTA DA neurons, which then release DA into the NAc[29]. Integrating these results with the aforementioned findings about insulin-mediated depression of excitatory synaptic transmission of VTA DA neurons in rodents[10], our data suggest that the peptide hormone insulin is a critical signal within this circuit in the human brain and suppresses salience response to food cues in the NAc as a consequence of inhibited drives from the VTA.

The palatability of food is decreased in fed compared to fasted states[32]. Insulin release from the pancreas following food intake and its subsequent action in the CNS appears to be an important modulator of this effect[10] and—in healthy individuals—might prevent short-term overconsumption of palatable food in environments in which food cues are ever-present. Interestingly, in individuals with normal fasting insulin levels, plasma insulin at baseline predicts food preference scores in the present study. This indicates that even though there is a large gradient between insulin concentrations in the blood and the cerebrospinal fluid[33], peripheral insulin is a good proxy for central insulin action under normal conditions.

Importantly, we also studied a group of non-diabetic, but insulin-resistant, individuals as identified by an established homeostatic model[34,35]. Insulin transport into the cerebrospinal fluid is thought to be attenuated in individuals with reduced whole-body insulin sensitivity[36] even though exact mechanisms are unknown and further factors can modulate insulin signalling in the brain (for example, genetic background)[25]. Investigating central insulin action might therefore be confounded by impaired transport of the hormone into the brain in individuals at risk for diabetes. Our approach of INI application overcomes this issue by delivering insulin rapidly along the olfactory nerves directly into the CNS, ensuring that only small amounts reach the systemic circulation and so do not acutely induce hypoglycaemia[24,37].

Our results from the baseline condition show lower food preference ratings in insulin-resistant individuals compared to participants with normal fasting insulin levels. This behavioural finding is mirrored by the specific decrease of food value signals in the NAc. According to the reward deficiency theory of obesity, individuals with lower sensitivity in DA-based reward regions tend to overeat as a means to compensate for decreased activation of these circuits. This theory is based on evidence that blocking DA D2 receptors increases appetite[17,18], findings that obese versus lean humans showed lower DA D2 (ref. 38) and $\mu$-opioid receptor availability in the striatum[39] and data demonstrating decreased striatal responses to food stimuli in obese individuals[40,41]. However, recent prospective findings[19] indicate that repeated overeating itself results in reduced striatal DA signalling[42] and reduced food preference[43]. Our neurobehavioural results converge with these findings by demonstrating specifically reduced responses to food-cues in individuals with IR, who have repeatedly eaten to excess in the past. Intriguingly, mice fed with a sweet high-fat meal

demonstrate depression of excitatory synaptic transmission in the VTA similar to effects observed after insulin induction and probably linked to elevated plasma insulin levels[10]. One may speculate that perpetually elevated levels of insulin in our insulin-resistant sample have led to chronically reduced modulation of mesolimbic pathways.

In contrast to insulin-sensitive individuals, insulin-resistant participants do not demonstrate decreased food value ratings after INI application, which is in accordance with central IR. Interestingly, this finding cannot be explained by body mass, which strengthens the validity of our grouping procedure and demonstrates that IR, but not necessarily obesity, is associated with insulin-induced changes in food valuation.

In line with behavioural findings, the optimal connectivity model in insulin-resistant individuals—but not in individuals with normal insulin levels—does not include an inhibitory modulatory input of INI on forward projections from the VTA to the NAc. This is in agreement with findings in a hyperinsulinemic mouse model suggesting that reduced insulin receptor efficacy in hyperinsulinemia reduces the capacity to cause a synaptic depression of VTA DA neurons by exogenous insulin induction[20].

Although insulin action in the brain of insulin-resistant participants is different to participants with normal insulin sensitivity, INI application induces some neurobehavioural changes in this group as well. Specifically, signals in the NAc increase and the optimal connectivity model in the IR group reveals a significant negative modulation of intrinsic self-connection by insulin in mesolimbic regions. Interestingly, neural patterns after INI together with observed trends in behaviour suggest that some reward signals tend to return to normal values, that is, those observed in the normal insulin-sensitive group at placebo. Thus, one may speculate that INI in these resistant individuals restores some reward deficits observed at baseline.

We observe complex group interactions in plasma insulin concentrations following the placebo compared to the insulin session. Although a slight dose-dependent permeation of INI into the circulation has been described before[37], differential spill-over effects between groups are unlikely given that no insulin-sensitive transporter/receptor is involved in the potential permeation of insulin into circulation. A more plausible explanation of this finding may be a complex interaction between our food paradigm and endogenous insulin metabolism in the insulin condition. This highly interesting question could be addressed in future studies, for example, by assessing c-peptide-levels after stimulation; this could provide information about endogenous insulin production.

Our modelling data suggest that the insulin-effects observed in the NAc are driven by insulin-action in the VTA; this extends the findings from animal research on insulin-mediated effects in the VTA to humans[10,11,20]. However, it is important to note that the effects of insulin on the striatum are probably more complex than what we have shown here. This is indicated by previous reports describing how insulin increases DA signalling via cholinergic interneurons in the NAc[12], which suggests regionally dependent roles of insulin (but also see[44]). Non-invasive fMRI data in humans only allow for indirect physiological conclusions and are strongly related to behavioural stimulation. The striking overlap of our behavioural and neural insulin findings with previous animal work[10,11,20,44], however, suggests that the underlying mechanisms are similarly comparable.

Of course, food-related behaviour involves a complex set of processes that include not only the evaluation of food palatability but also consummatory behaviour as well as the propensity to exert effort to obtain food. Interestingly, previous data in mice demonstrate that insulin decreased the salience for food-related cues only and did not mediate motivated behaviour, that is, insulin

did not alter the effort exerted to obtain palatable food[10]. In line with these results, hunger ratings in our study do not specifically change under insulin. Future experiments exploring motivated behaviour in more detail (for example, by using handgrip force as a motivational measure) may be able to further elucidate this aspect. In this context, there is ongoing debate over the precise role of the mesolimbic DA system for food-related reward aspects with more evidence indicating that DA does contribute to the incentive salience and valuation of stimuli but is less involved in the objective hedonic liking (for example, orofacial affective expression) for sensory pleasures[45,46]. In our study, we only obtained subjective preferences for food cues and the underlying process most likely reflects the salience and valuation of presented stimuli that typically is encoded in dopaminergic pathways[7,28] and that is an essential component within the reward circuitry[8]. In addition, an insulinergic modulation of the opioidergic pleasure system of the brain is possible given the intricate interconnections between the dopaminergic and opioidergic system in the NAc[47,48]. Combining objective hedonic liking assessments with opioidergic stimulation and recently established parcellation protocols on high resolution functional connectivity data in humans[49] might help to further disentangle multiple neurochemical modes within different NAc reward mechanisms.

The neural regulation of feeding behaviour in addition is thought to be modulated by other peptides, including leptin and ghrelin[2,50], as well as by prefrontally mediated self-control, which in turn seems to be sensitive to central insulin action[51]. It would also be interesting to investigate whether insulin has a more general effect on neurobehavioural responses to other primary reinforcers like sexual stimulation. The present approach offers a solid basis for targeting these aspects in future studies.

In conclusion, we provide data demonstrating that central insulin influences the valuation of food stimuli in humans, a finding that can be explained by the insulinergic modulation of mesolimbic pathways. Moreover, our results in insulin-resistant participants demonstrate the clinical relevance of an intranasal approach for assessing central insulin sensitivity and treating reward dysfunctions in individuals at risk for metabolic disorders.

## Methods

**Participants.** Forty-eight volunteers (20–34 years, $M = 25.8$, s.d. $= 3.3$; 25 female) participated in the present study after three individuals had been excluded since their fasting blood glucose levels (118, 113 and 122 md ml$^{-1}$) and eating protocols revealed that they did not follow the 10 h-fasting instruction. Participants were recruited via online announcements and existing databases. Exclusion criteria were current or previous psychiatric or neurological disorders, chronic and acute physical illness including diabetes, current psychopharmacological medication as well as MR-specific exclusion criteria. No participant followed any specific diet at the time of the experiment. To exclude systematic confounds during food evaluation, severe food allergies and adherence to a vegan diet constituted further exclusion criteria. All participants had normal or corrected-to-normal visual acuity.

As we were interested in participants with normal and aberrant insulin functioning, which is roughly correlated with body weight, 50% of our sample comprised lean adults (BMI 18.5–25 kg m$^{-2}$, $n = 24$), whereas the other age- and sex-matched half of our sample included overweight/obese participants (BMI 25.1–38 kg m$^{-2}$, $n = 24$). The local ethics committee approved the study and all participants gave written informed consent and were financially compensated for participation.

**Experimental protocol.** After successful screening, participants attended two experimental sessions, separated by at least 1 week. On each day, participants arrived in the morning between 7:30 and 10:30 h after an overnight fast of at least 10 h. After anthropometric measurements (see Table 1), ratings of feelings of current hunger and collection of blood samples (Fig. 1a, Assessment I), participants received 160 IU of insulin (Insuman Rapid, 100 IU ml$^{-1}$) or vehicle (0.27% m-Kresol, 1.6% glycerol, 98.13% water) by intranasal application. Participants received eight puffs per nostril, each puff consisting of 0.1 ml solution containing 10 IU human insulin or 0.1 ml placebo. The order of insulin and placebo was randomized and balanced, and the application was double blind. Before scanning, participants were familiarized with the task during a training session.

Participants began the preference paradigm (in the fMRI scanner) 30 min after INI was applied; this delay was introduced to ensure that the insulin had time to take effect[23]. After completion of the scans, participants again rated their feeling of hunger and a second set of blood samples was collected (Fig. 1a, Assessment II).

**Group classification.** Group definition was performed using the HOMA-IR, which has been widely employed in clinical research to assess insulin sensitivity and demonstrates high validity in non-diabetic samples[35]. The HOMA-IR-score was calculated based on fasting glucose and insulin concentrations derived from the samples before insulin/placebo was administered:

$$\text{HOMA} - \text{IR} = \frac{\text{Glucose}(\text{mmol}\,l^{-1})\,\text{x Insulin}(\mu U\,ml^{-1})}{22.5}$$

Participants with a score below 2 (ref. 26) on both scanning days were assigned to the NIR group.

**fMRI food-rating paradigm.** Two sets of stimuli were randomly presented on the two scanning days. Each one of the two parallel versions consisted of 70 food and 70 non-food colour images selected from the internet. All pictures had a size of $400 \times 400$ pixels and were presented on a white background. Food pictures featured both sweet and savoury items, with comparable amounts of high- and low-caloric items in every category. Pictures were specifically selected to cover common high- and low-palatable foods. Non-food pictures, such as accessories and trinkets (Fig. 1c), were chosen to evoke similar degrees of attractiveness. Validation of all sets was conducted in an independent sample ($n = 16$) and revealed that the two versions did not differ significantly regarding the mean preference ratings of the stimuli and average picture salience (Supplementary Table 1).

On each scanning day, food and non-food stimuli were pseudo-randomly presented (not more than three pictures from one category in a row) during three runs; each run lasted $\sim$12 min and runs were separated by a 1 min relaxation break (see Fig. 1b). Every run began with the instructions ('We will soon start with the question: Do you like the presented item or not?').

**MRI data acquisition.** All imaging data were acquired on a Siemens Trio 3T scanner (Erlangen, Germany) using a 32-channel head coil. Functional data were obtained using a multiband echo-planar imaging sequence. Each volume of the experimental data contained 60 slices (voxel size $1.5 \times 1.5 \times 1.5$ mm) and was oriented 30° steeper than the anterior to posterior commissure (AC–PC) line (repetition time (TR) = 2.26 s, echo time (TE) = 30 ms, flip angle = 80°, field of view (FoV) = 225 mm, multi-band mode, number of bands: 2).

An additional structural image (magnetization prepared rapid acquisition gradient echo (MPRAGE)) was acquired for functional preprocessing and anatomical overlay (240 slices, voxel size $1 \times 1 \times 1$ mm).

**fMRI data analysis.** Structural and functional data were analysed using SPM12 (Welcome Department of Cognitive Neurology, London, UK) and custom scripts in MATLAB. All functional volumes were corrected for rigid body motion and susceptibility artefacts (realign and unwarp). The individual structural T1 image was coregistered to the mean functional image generated during realignment. The functional images were spatially normalized and smoothed with a 4-mm full-width at half maximum isotropic Gaussian kernel.

A two-level random effects approach utilizing the GLM as implemented in SPM12 was used for statistical analyses. At the single subject level, onsets of food and non-food stimuli presentation were modelled as separate regressors convolving delta functions with a canonical hemodynamic response function. In addition, combined rating scores were entered as parametric modulators of food and non-food regressors separately. Data from the placebo and the insulin sessions were defined as separate sessions and entered into a single model. In all analyses, we accounted for the expected distribution of errors in the within-subject (dependency) and the between-group factors (unequal variance).

In the next step, we used the GLM denoise toolbox for Matlab[52] to improve signal-to-noise ratio in our data. Noise regressors were derived by conducting principal component and cross-validation analyses on voxel time-series that were identified, by an initial first-level-model fit, to be unrelated to the experimental paradigm. Individual noise regressors were then entered as regressors of no interest into the first-level model.

For each subject, contrast images for each regressor of interest were then entered into second-level random-effect ANOVA models including the factors stimulus (food/non-food), session (placebo/insulin) and group (NIR/IR).

We report results corrected for FWE due to multiple comparisons. We conduct this correction at the peak level within small volume ROI for which we had an a priori hypothesis or at the whole-brain level. Based on aforementioned central insulin findings in animals and the role in reward processing, we focused on the NAc and the VTA. To this end, we used functional ROIs (4 mm spheres) centred on the bilateralized peak voxels in the NAc ($\pm$12, 10, $-8$) and the VTA ($\pm$4, $-14$, $-12$) derived from 670 imaging studies on reward, as determined by a meta-analysis conducted on the neurosynth.org platform[53] (status September 2016, Fig. 4a).

**Dynamic causal modelling.** We used the DCM software implemented in SPM12 for effective connectivity analysis and a Bayesian model reduction approach[30]. The principal eigenvariate time-series were extracted from predefined unilateral masks of the NAc and the VTA (Fig. 4a), adjusted for effects of interest. To modulate effects of insulin in a single model, time series were then concatenated over the experimental days. A full DCM model was set up that was comprised of three factors: (i) the underlying connectivity architecture between the two regions, that is, extrinsic forward and backward connections between the VTA and the NAc, as well as intrinsic self-connections of the regions (fixed connections, A matrix), (ii) modulation of these connections by insulin (contextual modulation, B matrix) and (iii) visual stimuli as driving inputs into the nodes (exogenous inputs, C matrix). This full model was defined and inverted (estimated) for each participant. In total, the model space included 2,048 possible models: two driving inputs and four possible modulatory effects on four endogenous connections.

Next, a post-hoc model selection was used to create and test all possible reduced models in an unbiased way, while simultaneously reducing computational demand[54,55]. To identify the winner-model using Bayesian model selection at the group level, the evidence of each reduced model was pooled over all subjects within the group. With Bayesian parameter averaging, magnitudes and probabilities of each coupling parameter, as well as the magnitudes and effects with which the connections are modulated, were calculated[56]. Finally, we performed one-sample t-tests on the Bayesian parameter averages to determine which parameters differed significantly from zero. We repeated the procedure in each group separately for explorative reasons.

In addition to the Bayesian parameter averages across all participants (Fig. 6b), the post-hoc method also provides the single participants' individual parameters for the optimal model, which were extracted and entered into regression analyses including behavioural measures (Fig. 6d).

**Data availability.** Imaging data that support the findings of this study have been deposited online under the following link: http://neurovault.org/collections/JSYCRNOK/. Behavioural data are available at: https://figshare.com/s/16e8cea251ffec69cde1 (doi: 10.6084/m9.figshare.4987193).

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

## Acknowledgements

We thank V. Ott and M. Hallschmid for advice on the INI application setup as well as M. Tittgemeyer and J. Brüning for helpful discussions on the data. We gratefully acknowledge funding from the German Research Foundation (DFG, TR-SFB134).

## Author contributions

S.B., C.B. and L.T. designed the experiments. L.T., J.H., K.G. and P.F. performed the experiments. S.B., L.T. and S.M.S. analysed the data. S.B. and L.T. wrote the paper. All authors discussed the results and commented on the manuscript at all stages.

## Additional information

**Competing interests:** The authors declare no competing financial interests.

**Publisher's note**: 

