## [Peer Review File · Nature Communications]

Reviewers' comments:

Reviewer #1 (Remarks to the Author):

This was a very interesting study that examined the effects of intranasal insulin (INI) administration to insulin resistant and insulin sensitive individuals on mesolimbic activation during a food/non-food preference tasks. They found that in insulin sensitive individuals, INI depressed activation of the VTA and NAc during the food preference/palatability tasks. In contrast, in insulin resistant individuals had reduced food preferences compared to non-food items compared to non insulin resistant individuals. Insulin levels were inversely correlated with preference of food items, suggesting some evidence of 'reward deficits' related to hyperinsulinemia. Interestingly INI increased the right NAc BOLD response to food preference only in the insulin resistant group. Their dynamic causal modeling further indicated that the VTA to NAc connection was inhibited with INI. This is a very exciting paper. The novelty in these findings includes that 1) insulin resistance, but not necessarily obesity, is associated with decreased preference for food, 2) intranasal insulin decreases food value ratings in insulin sensitive but not insulin resistant individuals, and 3) intranasal insulin decreases mesolimbic circuit activation during the food preference tasks in insulin sensitive individuals, validating work in animal models. However, some of their interpretations based on previous work could use refining prior to publication.

1. In the introduction they interpret that because insulin decreases dopamine in the VTA are likely to underlie their observations that insulin reduces food valuation. However, it is not entirely clear what decreased somatodendritic dopamine would do. For example, decreased VTA dopamine may have less autoinhibitory effect on D2 receptors and could potentially increase firing of dopamine neurons. The stronger rationale may be that a decrease in glutamatergic synaptic transmission (LTD) may make it harder for VTA dopamine neurons to burst fire, thus likely decreasing dopamine into target regions (Labouebe et al., 2013 Nature Neurosci). Furthermore, hyperinsulinemic mice no longer exhibit this synaptic depression in the VTA (Liu et al., 2013 Nutr & Diabetes). Therefore I think they need to refine their interpretation of how somatodendritic dopamine contributes to these effects.

2. One of the major interpretations of the paper that they discuss is that INI to insulin resistant people restores the 'reward deficit' but increasing food liking scores. However, this data was only a trend ($P = 0.09$). They will need to tone down their discussion in this case, because arguably, this effect was due to chance.

3. Other work indicates that insulin can also regulate release of dopamine at terminals (Stouffer et al., 2015 Nat comm). This work shows that lower concentrations of insulin can increase dopamine release in the NAc and this effect is lost in obese mice. Thus, insulin may have a more complicated role at signaling energy balance in the mesolimbic system. These results should be discussed in the context of the current results.

Minor:

1. They should use their statistics program to illustrate the linear correlation in Fig 1e – it appears that the lines used were added after the fact as part of the blue line crosses the x axis.
2. The bar graph in figure d could be made taller.
3. Pg 17, In 341: In the discussion they indicate that “Since the excitatory connection from the VTA to the NAc was inhibited under intranasal insulin, dopaminergic drives of NAc value signals probably decreased in the insulin session ” – While I don't disagree with their interpretation, they don't have any evidence that this is an excitatory projection.
4. Pg 17, In 335: They incorrectly indicate that insulin mediates changes in motivated behaviour. This was assessed directly in Labouebe et al., 2013 and demonstrated that while intra-VTA insulin does not alter motivation or effort for food rewards, it did decrease the salience of cues and contexts associated with food.
5. In the discussion they bring up that in insulin resistant individuals, insulin may be less able to cross the blood brain barrier due to insulin receptor insensitivity and that INI bypasses this effect. It would be useful to have this information (supported appropriate with references) in the introduction to rationalize why INI would work in insulin resistant individuals. Is there evidence that the mesolimbic circuit less sensitive to insulin resistance?

Reviewer #2 (Remarks to the Author):

The authors of this paper studied the effect of intranasal insulin (INI) or placebo on food preference and brain activity in insulin resistant and insulin non-resistant subjects. They found that INI changed the hedonic value of food and modulated mesolimbic circuits. INI reduced food palatability ratings (food pictures) and value signals in mesolimbic areas in individuals with normal peripheral insulin sensitivity. In IR subjects food reward values returned to “normal” with INI.

Specific Issues:

The paper to be edited due to awkward sentences (eg lines 27-29) and some minor issues with grammar (eg lines 65-68).

The introduction of the paper summarizes the results of the paper. That should wait until the results section and discussion.

Line 95-99 are not clear. In one case the authors speak about normal sensitivity--then later use normal resistance and then peripheral insulin resistance. It is difficult to understand how they obtain their total numbers as well in this section. The methods sections is helpful

in understanding the subject numbers--but since the methods are discussed after the result section it would be helpful to clear up this issue earlier.

It is of interest that blood glucose and HbA1C did not differ between those with normal and abnormal insulin resistance. The authors might emphasize this and discuss possible reasons for this.

Were food preferences evaluated before choosing the foods? Perhaps IR and non-IR groups like different foods at baseline. This might influence the results of the paper

While significant, the changes in the preference scores of foods were small. Can those scores be verified as being biologically or behaviorally important?

What is $p < 0.05$ corrected?

There appears to be a problem with the sentence on lines 232-233.

In general, this is a very interesting paper. The authors simply need to edit the paper and clarify some of the methodological issues.

Reviewer #3 (Remarks to the Author):

Tiedemann and colleagues ran a mixed within- and between-subject study exploring the impact of intranasal insulin (INI) on brain and behavioral responses to food versus non-food images. These responses were compared across lean and overweight/obese people with the latter group showing relative insulin resistance. They observed that INI produced a reduction in food liking/preference in the lean group but a relative increase in the resistant group and that this interaction was associated with a reduction in mesolimbic responsivity in the former group. They conclude that this demonstrates how insulin modulates the "cross-talk between homeostatic and non-homeostatic feeding systems".

This is an interesting and very nicely designed and run study and the motivation and methods are set out clearly in a well-written paper. I do, however, have some concerns that the results are not really as easily interpretable in terms of direct effects of insulin as the authors suggest.

A primary concern is that the food and non-food items were not balanced for preference/liking. This makes it very hard to determine that the changes in liking ratings were specific to food or to rewarding items generally since, although the effects were seen as being specific to food items (compared to non-food) this analysis is potentially confounded by starting values. It is not difficult to find images that are properly matched (indeed, it's not clear how the stimuli were selected and matched for basic visual properties) and, without doing so, it's hard to relate the findings specifically to hedonic or non-hedonic eating as the authors seek to do.

Another concern is that the authors point out that it was only in the NIR group that the INI produced an increase in insulin levels but the table in supplementary material actually suggests that this interaction is driven in a complex way that is more difficult to interpret. Specifically, there was a significant decrease in insulin level in the IR group following INI, a decrease in the NIR following placebo but no significant increase in the NIR group following INI. This makes it hard to be clear on what precisely is going on here in respect of the insulin levels across the groups.

It is also important to note that hormone and metabolic profiles following a fast may well depend strongly on what was eaten the day before, prior to commencement of the fast. This is why in standard metabolic studies, a specific meal is given rather than just a generic instruction to fast for 10 hours – it's not clear that the current research has included this.

Finally, the INI has clearly had an effect on reward liking/preference in the IR group. It is difficult to be clear that the accompanying alteration in mesolimbic reactivity to the relevant cues is indeed a direct result of insulin on these regions or whether the reduced responses are a secondary effect of, say, reduced appetite such that the stimuli seem less desirable. This is important, since the authors are keen to interpret the insulin effects as being a direct modulation (as implied in the title) but it could easily be the case that it simply reduces motivation and that hedonic responses are attenuated as a downstream response to this. Ideally, it would have been a good idea to acquire data in both fasted and fed conditions since feeding represents a different means of reducing appetite/drive and, as such, could have been compared to insulin effects to allow a more precise interpretation of the latter.

Reviewer #4 (Remarks to the Author):

The present paper reports on the effect of intranasal insulin on food rating and activation of specific brain areas. The authors show that insulin action in the brain in lean, healthy subjects reduces ratings of food palatability. Furthermore, insulin action in the brain reduces food specific BOLD signals in the nucleus accumbens and the ventral tegmental area. In connectivity analyses, the authors found that insulin action in the brain inhibits projections from the ventral tegmental area to the nucleus accumbens. All the findings in lean healthy subjects are significantly different/attenuated from a group of overweight subjects, suggesting that those overweight subjects are insulin resistant in the brain

My specific comments are:

- 1) The grouping of the subjects is based on HOMA-IR. This is not a sufficient method to determine whole body insulin resistance, as it is based on a single blood fasting sample. Are there other parameters available (OGTT based insulin resistance (ISI Matsuda&deFronzo,) or clamp based insulin sensitivity index)
- 2) The grouping of the sample seems to me a fancy way of determining a lean versus overweight/obese group. If the statistics does not survive a correction for BMI, these are probably obesity-related effects (include insulin resistance) that just as well correlate with

BMI.

3) In the abstract and discussion, the authors state that in the "insulin resistant participants food reward deficits are normalized under INI. I think with this statement the authors relate to the findings shown in figure 4. In the normal weight group, brain insulin action suppresses food picture related activity in VAT and NAc. In the overweight group, brain insulin action increases food picture related activity in VAT and NAc. It is not correct to conclude that intranasal insulin "normalizes" deficits. In contrary, what we see here is that in overweight subjects insulin action in the brain is different to normal weight subjects. This is called insulin resistance of the brain. In overweight subjects, nasal insulin is not able to suppress/reduce suppresses food picture related activity in VAT and NAc. In these subjects the brain is "resistant" to insulin action in the brain in these areas.

This aspect should be pointed out in the discussion

4) The authors repeatedly use the term "hyposensitive subjects". Multiple recent studies investigating brain insulin action with intranasal insulin in obese individuals and patients with diabetes and dementia have coined the term "brain insulin resistant subjects". There are also numerous reviews available on this subject. I strongly suggest to interpret the findings in the framework of brain insulin sensitivity or resistance.

5) I do not understand figure 1 e. Here NIR show no correlation of food preference with insulin levels, whereas IR show an association. In the text, the opposite is stated. I hope there is only a mislabelling in figure 1e!?

6) On page 24/25 the authors mention that the NAc and the VTA are ROIs based on recent findings. Please indicate whether these are anatomical ROIs or functional? How was the exact MNI coordinate determined, based on what recent finding? Please add the appropriate citation.

7) In the introduction and the discussion, there is much speculation about dopamine signalling in the brain. Insulin may interact with dopamine signalling, and there is evidence from animal studies that this might be one mechanism explaining the findings of the present study. However, the present study is dealing with insulin signalling in the brain and does not allow to draw any mechanistic conclusions about dopamine signalling. So I suggest to shorten these speculative parts of the manuscript.

8) Author contribution (p26): for the second author, no contribution is listed

Minor points

7) An overview of the study design would be helpful in the main manuscript so that the reader can easily see at what time point fMRI, behavioral data and blood samples were collected.

8) At what time point was the post scan plasma insulin sampled? Was it the same for all participants? Did the insulin levels change from pre to post measurement. Where there any group differences?

9) On page 12, line 232, this not a complete sentence (see also comment 3 above)

Response to Reviewer #1

1. Need to refine interpretation of how somatodendritic dopamine contributes to the effects

We thank the Reviewer for emphasizing this very important aspect and for pointing us to the paper of Liu and colleagues which nicely fits with our findings in insulin-resistant participants. We now extended our discussion to cover potential mechanisms underlying insulin-mediated effects in mesolimbic brain regions in the introduction and the discussion. We agree that findings from Labouebe and colleagues provide a strong rationale for the observed effects.

Introduction

Page 3, line 41ff.:

For example, direct administration of insulin into the VTA reduces hedonic feeding under sated conditions and depresses somatodendritic DA in the VTA. Insulin-induced depression of somatodendritic dopamine has been attributed to the upregulation of the number or function of DA transporter in the VTA (Mebel et al., 2012). Moreover, insulin injection decreases glutamatergic synaptic transmission (long-term depression, LTD) onto VTA dopamine neurons which in turn may reduce dopamine burst activity and subsequent DA release in mesocorticolimbic dopaminergic regions (Labouebe et al., 2013).

Page 4, line 66ff.:

The potential role of elevated central insulin levels on the regulation of reward signals in hyperinsulinemic humans, however, have not been studied so far. In this context, interesting findings in rodents have demonstrated that sweetened high fat exposure induces synaptic depression onto dopamine neurons (Labouebe et al., 2013) and that insulin-mediated LTD of VTA dopamine neurons is reduced in hyperinsulinemia (Liu et al., 2013).

Discussion

Page 18, line 359ff.:

These findings validate and extend work in animal models that demonstrated insulin-mediated depression of DA activity in the VTA paralleled by decreased hedonic processing of food stimuli (Labouebe et al., 2013, Mebel et al., 2012, Könnner et al., 2011). For example, insulin injection in rodents induced long-term depression (LTD) of excitatory synapses onto VTA dopamine neurons which probably attenuates dopamine release in the mesocorticolimbic dopamine system and selectively reduced the preference for contextual cues associated with food reinforcement, as measured by conditioned place preference (Labouebe et al., 2013).

Page 19, line 378ff.:

Integrating these results with aforementioned findings of insulin-mediated depression of excitatory synaptic transmission of VTA dopamine neurons (Labouebe et al., 2013), our data suggest that the peptide hormone insulin is a critical signal in this circuit and suppresses reward response to food cues in the NAc as a consequence of inhibited drives from the VTA.

Page 20, 410ff.:

Intriguingly, mice fed with a sweet high fat meal demonstrate depression of excitatory synaptic transmission in the VTA similar to effects observed after insulin induction and probably linked to elevated plasma insulin levels (Labouebe et al., 2013). One may speculate that perpetually elevated levels of insulin in our insulin-resistant sample have led to chronically reduced modulation of mesolimbic pathways.

Page 20, line 422ff.:

This is in agreement with findings in a hyperinsulinemic mouse model suggesting that reduced insulin receptor efficacy in hyperinsulinemia reduces the capacity to cause a synaptic depression of VTA dopamine neurons by exogenous insulin induction (Liu et al., 2013).

2. The interpretation of behavioural effects in insulin-resistant individuals

We agree that we need to tone down the interpretation and we revised the manuscript w.r.t. this aspect.

Abstract, line 22ff.:

In a group of insulin-resistant participants, we observed food-reward deficits at baseline and aberrant central insulin action.

Page 21, line 426ff.:

Although insulin action in the brain of insulin-resistant participants was different to participants with normal insulin sensitivity, INI application induced some neurobehavioural changes in this group. Specifically, signals in the NAc increased and the optimal connectivity model in the IR group revealed a significant negative modulation of intrinsic self-connection by insulin in mesolimbic regions. Interestingly, neural patterns under insulin together with observed trends in behaviour (Fig. 3a) suggest that some reward signals tended to return to “normal” values, i.e. those observed in the normal insulin-sensitive group at placebo. Thus, one may speculate that INI in these resistant individuals restores some “reward deficits” observed at baseline.

3. Complex role of central insulin

Thank you for mentioning this important aspect. We now refer to the work of Stouffer and colleagues in the discussion:

Page 21, line 444ff.:

Our modelling data suggest observed insulin-effects in the NAc to be driven by insulin-action in the VTA which transfers work in animal models on insulin-mediated effects in the VTA to humans (Labouebe et al., 2013, Mebel et al., 2012, Liu et al., 2013). However, it is important to note that insulin effects in the striatum are probably more complex than this. This becomes obvious by previous reports about insulin effects on increasing DA signalling in the NAc mediated by cholinergic interneurons (Stouffer et al., 2015, but also see Schoffemeer et al., 2011) which suggests regionally dependent

roles of insulin. Non-invasive fMRI data in humans only allow for indirect physiological conclusions and are strongly related to behavioural stimulation. The striking overlap of our behavioural and neural insulin findings with previous animal work (Labouebe et al., 2013, Mebel et al., 2012, Liu et al., 2013, Schoffemeer et al., 2011), however, argues for a similar overlap regarding underlying mechanisms.

4. Regression line Fig. 1e

...has been corrected.

5. Format bar graphs

...was revised.

6. Wording excitatory connection

We agree that this wording might be misleading here and replaced “excitatory” with “positive”.

7. Insulin effects and motivated behaviour

We apologize for this mistake. We corrected that and also added the following:

Page 22, line 453ff.:

Of course, food-related behaviour involves a complex set of processes that includes not only the evaluation of food palatability but also consummatory behaviour as well as the propensity to exert effort in order to obtain food. Interestingly, previous data in mice demonstrate that insulin decreased only the salience for food-related cues while it did not mediate motivated behaviour, i.e. insulin did not alter the effort exerted to obtain palatable food (Labouebe et al., 2013). In this line, in our study hunger ratings did not specifically change under insulin. Future experiments exploring motivated behaviour in more detail (e.g. by using handgrip force as a motivational measure) may be able to further elucidate this aspect.

8. INI and insulin resistance

According to the Reviewer's suggestion, we have added the following paragraph to the introduction:

Page 4, line 74ff.:

To study central insulin effects under physiological and pathological circumstances we investigated participants with normal insulin sensitivity as well as non-diabetic individuals with insulin resistance, who are at risk for T2D (Kahn et al., 2000, Shanik et al., 2008). In a placebo-controlled double-blind cross-over design, central insulin effects were investigated by making use of the intranasal route of insulin administration (INI). INI application has been shown in humans to bypass the blood-brain-barrier (BBB) and effectively deliver insulin to the CNS within 30 minutes after administration in the absence of relevant systemic absorption (Born et al., 2002, Spetter et al., 2015). By using this approach we are able to rule out that our findings in individuals with reduced whole-body insulin sensitivity are confounded by potentially attenuated transport of the hormone across the BBB (Heni et al., 2015).

Response to Reviewer #2

1. Paper editing

We apologize for any errors that may have slipped through in the process of editing the paper. We thoroughly edited the revised manuscript.

2. Summary in the introduction

...has been removed.

3. Terminology insulin resistance and recruitment details

We agree that some of our terminology may have been confusing. According to the Reviewer's suggestion we revised the paragraph on study groups in the result section:

Page 6, line 115ff.:

Forty-eight normal to overweight non-diabetic volunteers participated in the study and were classified into insulin groups based on insulin sensitivity as defined by the well-established homeostatic model assessment using a cut-off of < 2.0 (HOMA-IR) (Gayoso-Diz et al., 2013). Normal insulin sensitivity was identified in $N = 28$ participants (NIR; 14 male), while $N = 20$ individuals fulfilled criteria for insulin resistance (IR; 9 male).

Moreover, we homogenized the wording regarding group labelling throughout the manuscript, i.e. used the labelling "normal insulin sensitivity" versus "insulin resistance".

4. Blood glucose and HbA1C

This is a very important aspect and we thank the Reviewer for mentioning this. Indeed, diabetes, as indicated by an elevated HbA1C value, was a strict exclusion criterion in the current study. We were particularly interested in non-diabetic individuals with reduced peripheral insulin sensitivity, i.e., insulin resistance, who are at risk for T2D (Page 4, line 74ff.).

Normal values in HbA1C and fasting glucose validate this inclusion criterion and argue for a still successful compensation of insulin resistance as expected in a non-diabetic risk-group, i.e. beta cells are still able to produce enough insulin to overcome insulin resistance and

keep blood glucose levels in the normal range. We included this line of argumentation into the manuscript:

Page 6, line 119ff.:

Normal HbA1C values confirm the exclusion of diabetes in our insulin-resistant participants who are at risk for T2D but in whom elevated insulin release may still compensate for reduced insulin sensitivity (Table 1).

5. General food preferences

This is an interesting question. First of all, in order to exclude systematic confounds during food evaluation, severe food allergies (e.g. nuts) and adherence to a vegan diet constituted exclusion criteria and no participant followed any specific diet (Page 23, line 482ff.). In addition, we chose a wide range of different food categories. However, based on the Reviewer’s question we ran an additional analysis in which we compared the general liking of different food categories between groups at baseline. Specifically, we classified all food stimuli into sweets, salty snacks, dairy products, fast-food, fruits, baked goods, tapas, and vegetables. We then ran Chi-Square tests to analyse potential group differences in the general liking (yes vs. no) of different food categories. Results revealed no significant differences (all $p > .20$). We added this sub-analysis to the supplement (Supplementary Table 2) and refer to this finding in the main text:

Page 6, line 122f:

General preference for different kind of foods was comparable in both groups (Supplementary Table 2).

	Food categories (% liked/total)							
	Sweets	Salty snacks	Dairy products	Fast-food	Fruits	Baked goods	Tapas	Vegetables
NIR	.92 (.05)	.82 (.07)	.96 (.04)	.82 (.07)	.89 (.06)	.89 (.06)	.50 (.09)	.92 (.05)
IR	.90 (.07)	.85 (.08)	.85 (.08)	.90 (.07)	.95 (.05)	.85 (.08)	.65 (.11)	.90 (.07)
p	n.s.	n.s.	n.s.	n.s.	n.s.	n.s.	n.s.	n.s.

Table | General food preferences

6. Biological and behavioural impact of findings

On average ~5% less food stimuli were liked (yes versus no) under insulin in the normal insulin-sensitive group which corresponds to ~4 food items. The mode of the distribution was 10% and individual values raised up to 27%. Twenty out of 28 participants with normal insulin sensitivity demonstrated an obvious decline in food palatability ratings under insulin. The palatability of food is decreased in fed compared to fasted state (e.g. Cameron et al., 2014, PlosOne) which suggests this marker to be relevant in the context of metabolic state in healthy individuals (please also see Page 19, line 382ff.). Biologically, Labouebe and colleagues (2013, Nature Neuroscience) report long-term depression of VTA dopamine neurons also in mice fed with a sweetened high-fat meal together with increased plasma insulin levels and reduced cocaine-reduced locomotor activity that is similar to the effects they observed after insulin induction. Based on these data we believe that our significant findings are highly relevant on the behavioural and the biological level by validating insulin to be a key signal in feeding relevant neurocircuits and behaviours.

7. P<.05 corrected

The expression $p < .05$ corrected refers to the family wise error (FWE) correction of the imaging data as described on Page 11, line 222ff. We agree that using only “corrected” may be misleading here and replaced “ $p < .05$ corrected” with “ $p < .05$ FWE corrected” throughout the manuscript.

1. Balance of food and non-food items

Unfortunately, it appears to be almost impossible to find depicted stimuli that are as liked as pictures of the primary reinforcer food across a wide range of individuals. This holds especially true when trying to avoid subgroup confounds, i.e. erotic stimuli may evoke similarly high likability ratings in men but there is a strong confounding gender effect. Since we wanted to maximize the generalizability of our findings by investigating a representative sample (men and women) we had to exclude such stimuli.

In general, based on the established role of insulin on food processing our hypotheses focused on effects on food-stimuli processing while non-food items were primarily presented to control for general task demands and baseline effects. Specifically, by using differential contrasts within subjects (i.e. food > non-food) in our behavioural and imaging analyses we were able to control for potential confounding effects by motor responses, visual processing, general evaluation processes etc. Moreover, our fMRI results mainly focus on parametric analyses, i.e. on modulation by likeability ratings which reveal brain regions that linearly modulate likeability while mean effects (i.e. intercept) are not considered. In this context it is important to note that both food and non-food stimulus ratings covered the whole range of likeability values (1-8), and there was no difference in variability (Levene Test, $F = 2.3$, $p > .12$).

However, to address the Reviewer's concern in more detail, we ran a control analysis on stimuli subsets in which the preference of food and non-food stimuli was matched based on the liking values observed in the normal insulin-sensitive group at baseline using an iterative item-selection approach (cut-off food vs. non-food $p > .05$). This resulted in 38 food and 38 non-food items which did not differ w.r.t. preference scores in the NIR ($T_{(27)} = 1.661$, $p = .11$) and also not in the IR individuals ($T_{(19)} = -.59$, $p = .56$). Most importantly, even in these reduced sets we observed a food specific insulin effect in the NIR group ($F_{(1, 27)} = 4.395$, $p = .046$). This effect was also still significant compared to the IR group ($F_{(1, 46)} = 4.291$, $p = .044$), indicating that insulin mediates food but not non-food preference in stimuli matched for likeability. Results of this sub-analysis are displayed in the following figure:

Figure | Behavioural results for the matched stimulus sets.

Nevertheless, we agree with the Reviewer that our wording w.r.t. specificity might be misleading in this context, since we do not know whether insulin may have a more general impact on the processing of other highly hedonic stimuli (e.g. erotic pictures). We thus revised the terminology throughout the manuscript and included speculations about general insulin effects on hedonic networks in the discussion:

Page 22, line 465ff.:

In this context, it would also be interesting to investigate whether insulin has a more general effect on neurobehavioural responses to other primary reinforcers like sexual stimulation. The present approach offers a solid basis for targeting these aspects in future studies.

Regarding the procedure of our picture search we primarily focused on including a large range of likeability as well as similar visual feature complexity, contrast, brightness and composition, e.g., all stimuli (food and non-food) had a white background and were depicted without any irrelevant details (e.g. package). Consequently, stimulus sets did not differ w.r.t. picture salience (Supplementary Table 1).

2. Plasma insulin effects

We apologize for the confusion caused by our misleading statement. What we actually see, except for the insulin condition in normal subjects, is an expected (trend to) decrease of

insulin over time (pre- vs. post-scanning) due to further fasting during the study. This decrease in the NIR group was significantly stronger during PL compared to IN. We corrected the notion of this complex finding in the results and the discussion:

Page 7, line 139ff.:

As expected, plasma insulin levels across all participants decreased over time ($F_{(1,46)} = 8.16$; $p = .006$) and this decrease across individuals did not differ between the insulin and the placebo session ($F_{(1,46)} = 1.25$; $p = .27$). There was however a significant group by session interaction ($F_{(1,46)} = 4.44$; $p = .04$) which mainly was driven by a stronger plasma insulin decline at the placebo compared to the insulin session in NIR-individuals (see Supplementary Table 4).

Page 21, line 435ff.:

We observed complex group interactions in plasma insulin concentrations following the placebo compared to the insulin session. Although a slight dose-dependent permeation of INI into the circulation has been described before (Ott et al., 2015), differential spill-over effects between groups are unlikely given that no insulin-sensitive transporter/receptor is involved in the potential permeation of insulin into the circulation. A more plausible explanation of this finding may be a complex interaction between our food paradigm and endogenous insulin metabolism in the insulin session. This highly interesting question could be addressed in future studies, for example by assessing c-peptide-levels after stimulation that could provide information on endogenous insulin production.

3. Standardized meal

We agree that it would have been ideal to have had a standard meal but due to logistic reasons (e.g. monitoring of meal intake; comparable likeability of the meal in all participants) this was not feasible. Our participants fasted for more than 10 hours (mean fasting time at both days 12:45 hours) and fasting glucose levels (Table 1) confirmed fasting state in all participants. Fasting time and also hunger ratings did not differ between groups, between placebo and insulin day nor were there any group x session interactions (all $p > .33$, please also see Page 7, line 135ff.). The long fasting time made us confident that potential effects

due to metabolic differences caused by last meals are unlikely in our non-diabetic cohort. In this line, previous findings in large data sets demonstrated no relevant further impact on fasting blood glucose after a fasting duration of 8 hours (Moebus et al., 2011: Impact of time since last caloric intake on blood glucose levels; European Journal of Epidemiology).

Based on the Reviewer’s concern and to further rule out a systematic impact of last meals, we ran an analysis on the last caloric intake that was protocolled in all participants for both study days. Computation of caloric intake was conducted using the software DGExpert 1.8.6 (German Nutrition Society).

This analysis demonstrate no significant group effect in caloric intake on the placebo ($T_{(46)} = 1.3$; $p > .20$) or the insulin day ($T_{(46)} = .94$; $p > .36$), no significant effect of placebo vs. insulin day ($T_{(47)} = .74$; $p > .46$) and no significant day x group interaction ($F_{(1,46)} = .06$, $p > .81$). Since these findings strengthen our assumptions on comparable fasted states we included them into the Supplement and refer to them in the main text:

Page 7, line 127ff.:

Additional analyses on the caloric content of the protocolled last meal prior to fasting in each participant demonstrated no difference in caloric intake between sessions, groups and regarding group x session interactions (Supplementary Table 3a).

a

	Prior food intake (kcal)		p	p (NIR vs. IR)	p (group x session)
	PL	IN			
NIR	329.10 (31.28)	302.42 (27.42)	n.s.		
IR	274.92 (22.66)	262.09 (33.66)	n.s.	n.s.	n.s.

b

	Fasting duration (hours)		p	p (NIR vs. IR)	p (group x session)
	PL	IN			
NIR	12.78 (.31)	12.75 (.28)	n.s.		
IR	12.72 (.24)	13.08 (.29)	n.s.	n.s.	n.s.

Table | (a) Last caloric intake and (b) fasting duration before each study day.

4. Insulin effects on motivation

We agree with the Reviewer that an impact of motivated behaviour on our results cannot be ruled out by our design. However, there are several reasons why we think hedonic food-valuation is directly modulated by insulin and not simply a consequence of changes in appetite in our study:

1. Feelings of hunger were recorded on both study days before and after scanning in all participants and did not show any group, session or group x session effects (all $p > .2$) nor were differences in hunger ratings directly correlated with insulin effects on food palatability ($r = .20$, $p > .31$).
2. This indirectly fits with interesting findings of Labouebe and colleagues (Nature Neuroscience, 2013) who found insulin action in the VTA to decrease the salience for food-related cues while they did not find insulin to mediate motivated behaviour, i.e. insulin did not alter the effort exerted to obtain palatable food.
3. Changes in hedonic values of food were directly related to mesolimbic findings in our study.

We added these aspects into the discussion:

Page 22, line 453ff.:

Of course, food-related behaviour involves a complex set of processes that includes not only the evaluation of food palatability but also consummatory behaviour as well as the propensity to exert effort in order to obtain food. Interestingly, previous data in mice demonstrate that insulin decreased only the salience for food-related cues while it did not mediate motivated behaviour, i.e. insulin did not alter the effort exerted to obtain palatable food (Labouebe et al., 2013). In this line, in our study hunger ratings did not specifically change under insulin. Future experiments exploring motivated behaviour in more detail (e.g. by using handgrip force as a motivational measure) may be able to further elucidate this aspect.

We agree that, ideally, a further condition “fed state” would have allowed for addressing very interesting questions, even though this would have massively increased the logistic costs of this study. Most importantly, it would have been hard to disentangle neural findings

in our sample of insulin-resistant participants in whom both potentially altered peripheral insulin response to food intake as well as potentially reduced transport of insulin into the brain (Page 5, line 79ff., Page 19, line 391ff.) might have impacted on central findings. Our approach of studying central insulin function by means of intranasal insulin application overcomes these issues.

Response to Reviewer #4

1./2. HOMA-IR and BMI

We agree that OGTT or clamp based insulin sensitivity assessment would have been ideal in this study. However, there is large literature demonstrating the high validity (e.g. reported correlations of HOMA-IR with clamp assessment $\sim .88$) of the frequently used HOMA-IR in non-diabetic participants (reviewed in Wallace et al., Diabetes Care, 2004). In this context, we thank the Reviewer for recommending the consideration of BMI as a potential confound of our results. Based on these comments we re-ran all our behavioural analyses and found BMI to not influence our main findings (see below). In our opinion, this finding strikingly strengthens the validity of our grouping procedure by demonstrating that insulin resistance, but not necessarily obesity is associated with insulin-induced changes in hedonic food valuation. We are happy to provide and discuss these aspects in the revised manuscript:

Page 10, line 185ff.:

In order to rule out that observed changes were primarily driven by differences in body weight we re-ran the behavioural analyses including BMI as a covariate. Results revealed no significant impact of BMI on observed changes, i.e. group interactions in insulin effects remained significant ($F_{(1,45)} = 6.50$; $p = .014$). Moreover, there was no significant correlation between BMI and insulin-mediated changes in food preference scores across participants ($r = -.11$; $p = .46$).

Page 20, line 417ff.:

Interestingly, this finding could not be explained by body mass which strengthens the validity of our grouping procedure and demonstrates insulin resistance but not necessarily obesity, to be associated with insulin-induced changes in hedonic food valuation.

3. /4. Insulin resistance

We apologize for any confusion caused by our wording regarding insulin resistance in the original manuscript. According to the Reviewer's suggestion we carefully revised the

manuscript consistently using the recommended terminology “normal insulin-sensitive” and “insulin resistance”.

Page 20, line 415ff.:

In contrast to insulin-sensitive individuals, insulin-resistant participants did not demonstrate decreased food value ratings after INI application which is in accordance with central insulin resistance.

We also agree that, since this data was only a trend ($p = .09$) we have to tone down the discussion of potential normalization in the insulin-resistant group. To this end, we revised this part in the abstract.

Abstract, line 22ff.:

In a group of insulin-resistant participants, we observed food-reward deficits at baseline and aberrant central insulin action.

In addition, we indicate more clearly that we can only speculate about potential normalization patterns in the insulin-resistant group:

Page 21, line 426ff.:

Although insulin action in the brain of insulin-resistant participants was different to participants with normal insulin sensitivity, INI application induced some neurobehavioural changes in this group. Specifically, signals in the NAc increased and the optimal connectivity model in the IR group revealed a significant negative modulation of intrinsic self-connection by insulin in mesolimbic regions. Interestingly, neural patterns under insulin together with observed trends in behaviour (Fig. 3a) suggest that some reward signals tended to return to “normal” values, i.e. those observed in the normal insulin-sensitive group at placebo. Thus, one may speculate that INI in these resistant individuals restores some “reward deficits” observed at baseline.

As far as we know, this is the first study that assesses central insulin action in mesolimbic reward circuits during food-relevant behaviour in participants with defined insulin

resistance. There is an ongoing discussion about how both peripheral and central insulin resistance are related to each other, e.g. where insulin resistance starts and whether it can be overcome by insulin treatment (Heni et al., Nature Reviews Endocrinology, 2015). Our first data may provide a starting point for assessing potential restoring effects of exogenous insulin application, e.g. by using different dosages and treatment durations.

5. Figure 1e

We apologize for this mistake! There was indeed a mislabelling in Figure 1e, that has been corrected (now Fig. 2c).

6. Regions of interest

We used functional ROIs (4-mm spheres) centred on the bilateralized peak voxels in the NAc (-12, 10, -8) and the VTA (4, -14, -12) resulting from 670 imaging studies on “reward” meta-analysed on the neurosynth.org platform (Yarkoni et al., Nature Methods, 2011; status September 2016, see Figure 4a). We added these details into the revised manuscript.

Page 26, line 565ff.:

To this end, we used functional ROIs (4-mm spheres) centred on the bilateralized peak voxels in the NAc ($\pm 12, 10, -8$) and the VTA ($\pm 4, -14, -12$) derived from 670 imaging studies on “reward” meta-analysed on the neurosynth.org platform (Yarkoni et al., 2011, status September 2016, Fig. 4a).

7. Dopamine

Of course we can only speculate about dopamine effects in our fMRI study. We further made clear that – even though there is a strong overlap with behavioural and neural findings from animal studies tracking insulin and dopamine effects – our method only allows for indirect conclusions:

Page 21, line 444ff.:

Our modelling data suggest observed insulin-effects in the NAc to be driven by insulin-action in the VTA which transfers work in animal models on insulin-mediated effects in the VTA to humans (Labouebe et al., 2013, Mebel et al., 2012, Liu et al., 2013).

However, it is important to note that insulin effects in the striatum are probably more complex than this. This becomes obvious by previous reports about insulin effects on increasing DA signalling in the NAc mediated by cholinergic interneurons (Stouffer et al., 2015, but also see Schoffemeer et al., 2011) which suggests regionally dependent roles of insulin. Non-invasive fMRI data in humans only allow for indirect physiological conclusions and are strongly related to behavioural stimulation. The striking overlap of our behavioural and neural insulin findings with previous animal work (Labouebe et al., 2013, Mebel et al., 2012, Liu et al., 2013, Schoffemeer et al., 2011), however, argues for a similar overlap regarding underlying mechanisms.

8. Author contribution

We added further details on author contribution.

9. Overview of the study design

... was added in Figure 1.

10. Insulin sampling

We used the same timing in all participants. The exact timing is now indicated in the study roadmap in Figure 1. Changes in insulin levels are reported in Supplementary Table 4 and mentioned on Page 7, line 135ff. and on Page 21, line 435ff. Most importantly, plasma insulin changes were not correlated with reductions in hedonic food values:

Page 10, line 192ff.:

We then investigated whether potential plasma insulin changes during the insulin session were directly associated with observed behavioural insulin effects (i.e. score changes in food liking) and found no significant correlations across and within groups (all $p > .21$).

11. Incomplete sentence

... has been corrected.

REVIEWERS' COMMENTS:

Reviewer #1 (Remarks to the Author):

I think revised manuscript is very interesting, novel and addresses an important knowledge gap in the field. The authors have addressed all of my previous concerns.

I do have one final issue however with the interpretation that mesolimbic dopamine is mediating hedonic aspects of feeding. (For example; Line 38: The mesolimbic pathway is critically involved in mediating rewarding effects of both food and drugs and several other places throughout the text) . I know that this is written over and over in other publications and in popular media, but it is inaccurate. It has been known for a long time that inhibition of dopamine receptor signaling or dopaminergic lesions do not block affective 'hedonic' responses to food intake (see K. Berridge work), nor does enhancing dopamine increase the palatability of food. A review I highly recommend they read prior to revising some of the statements in their manuscript is "Dopamine and food addiction: a lexicon badly needed" (Salamone Correa 2013 Biol Psych PMID: 23177385). It is likely that dopamine is more involved in the salience of the food or cues/context associated with the food, value and action to obtain food rather than the 'hedonic impact' of food. For example, because individuals in the study rated liking of food vs non-food pictures, this may reflect the salience of the food representation (cue) rather than the presumed palatability/affective response to food itself. If updated to reflect the current roles of the mesolimbic system in ingestive behavior, the paper would be much stronger. I don't think these changes the novelty and high interest of their findings.

Reviewer #2 (Remarks to the Author):

The authors have responded adequately to the questions that were raised. I believe this manuscript is now acceptable for publication.

Reviewer #3 (Remarks to the Author):

I thank the authors for responding to my review thoroughly and positively. I think they have done everything possible to address my concerns.

In some instances the concerns persist despite clever reanalyses of the existing data. But in addition the authors have been open and sincere in acknowledging outstanding ambiguities.

Overall, I feel that this is an interesting paper and that it makes a valuable contribution to the field. Therefore, despite persistent concerns about the afore-mentioned ambiguities, I would support its publication and congratulate the authors on their achievements.

Reviewer #4 (Remarks to the Author):

I am still not satisfied with the discussion, page 19, line 390-397.

Insulin resistance of the brain is here described exclusively as a problem of reduced transport of insulin to the brain (altered BBB). However, there are various alternative possibilities of impaired insulin signalling in the brain (genetic, epigenetic etc). This should be mentioned.

Furthermore, the last 6 words of line 397, page 19 are not correct. There is evidence that insulin action in the brain is indeed altering systemic glucose homeostasis (Heni et al Diabetes 2014 and 2017). This should be corrected.

Comments reviewer #1:

I think revised manuscript is very interesting, novel and addresses an important knowledge gap in the field. The authors have addressed all of my previous concerns.

I do have one final issue however with the interpretation that mesolimbic dopamine is mediating hedonic aspects of feeding. (For example; Line 38: The mesolimbic pathway is critically involved in mediating rewarding effects of both food and drugs and several other places throughout the text) . I know that this is written over and over in other publications and in popular media, but it is inaccurate. It has been known for a long time that inhibition of dopamine receptor signaling or dopaminergic lesions do not block affective 'hedonic' responses to food intake (see K. Berridge work), nor does enhancing dopamine increase the palatability of food. A review I highly recommend they read prior to revising some of the statements in their manuscript is "Dopamine and food addiction: a lexicon badly needed" (Salamone Correa 2013 Biol Psych PMID: 23177385). It is likely that dopamine is more involved in the salience of the food or cues/context associated with the food, value and

action to obtain food rather than the 'hedonic impact' of food. For example, because individuals in the study rated liking of food vs non-food pictures, this may reflect the salience of the food representation (cue) rather than the presumed palatability/affective response to food itself. If updated to reflect the current roles of the mesolimbic system in ingestive behavior, the paper would be much stronger. I don't think these changes the novelty and high interest of their findings.

Response:

We further restricted the description and interpretation of our behavioural measurement to the valuation of food (cues) and carefully revised our terminology concerning the terms 'hedonic' and 'reward' throughout the manuscript. In addition and as recommended by the reviewer, we added a new paragraph into the discussion that further refers to the currently discussed role of the mesolimbic system in ingestive behaviour (page 23):

"In this context, there is ongoing debate over the precise role of the mesolimbic dopamine system for food-related reward aspects with more evidence indicating that dopamine does contribute to the incentive salience and valuation of stimuli but is less involved in the objective hedonic liking (e.g. orofacial affective expression) for sensory pleasures (Berridge, 2007; Salamone & Correa, 2013). In our study, we only obtained subjective preferences for food cues and the underlying process most likely reflects the salience and valuation of presented stimuli that typically is encoded in dopaminergic pathways (Haber & Knutson, 2010; Peters & Büchel, 2010) and that is an essential component within the reward circuitry (Berridge & Kringelbach, 2015). Additionally, an insulineric modulation of the opioidergic pleasure system of the brain is possible given the intricate interconnections between the dopaminergic and opioidergic system in the NAc (Castro & Berridge, 2014, Tuominen et al., 2015). Combining objective hedonic liking assessments with opioidergic stimulation and recently established parcellation protocols on high resolution functional connectivity data in humans (Baliki et al., 2013) might help to further disentangle multiple neurochemical modes within different NAc reward mechanisms."

Comments reviewer #4:

I am still not satisfied with the discussion, page 19, line 390-397.

Insulin resistance of the brain is here described exclusively as a problem of reduced transport of insulin to the brain (altered BBB). However, there are various alternative possibilities of impaired insulin signalling in the brain (genetic, epigenetic etc). This should be mentioned.

Furthermore, the last 6 words of line 397, page 19 are not correct. There is evidence that insulin action in the brain is indeed altering systemic glucose homeostasis (Heni et al Diabetes 2014 and 2017). This should be corrected.

Response:

We revised the mentioned section accordingly (pages 19f):

“Insulin transport into the CSF is thought to be attenuated in individuals with reduced whole-body insulin sensitivity (Heni et al., 2014) even though exact mechanisms are unknown and further factors can modulate insulin signalling in the brain (e.g. genetic background) (Heni et al., 2015). [...] Our approach of intranasal insulin application overcomes this issue by delivering insulin rapidly along the olfactory nerves directly into the CNS, ensuring that only small amounts reach the systemic circulation and so do not acutely induce hypoglycaemia (Spetter & Hallschmid, 2015; Ott et al., 2015).“